# Pregnancy and Neonatal Outcomes after Transfer of Mosaic Embryos: A Review

**DOI:** 10.3390/jcm10071369

**Published:** 2021-03-27

**Authors:** Sina Abhari, Jennifer F. Kawwass

**Affiliations:** Division of Reproductive Endocrinology and Infertility, Department of Gynecology and Obstetrics, Emory University School of Medicine, Atlanta, GA 30322, USA; Jennifer.kawwass@emory.edu

**Keywords:** IVF, PGT-A, mosaic embryo, segmental mosaicism, whole chromosome mosaicism, pregnancy outcomes, neonatal outcomes

## Abstract

Preimplantation genetic testing for aneuploidy (PGT-A) seeks to identify embryos with a normal chromosome complement during in vitro fertilization (IVF). Transfer of one euploid embryo at a time maximizes the chance of implantation while minimizing the risk of multiple pregnancy. The emergence of new technologies including next generation sequencing (NGS) has led to increased diagnosis of embryonic mosaicism, suggesting the presence of karyotypically distinct cells within a single trophectoderm (TE). Clinical implications of embryonic mosaicism are important in both naturally conceived and IVF pregnancies. Although information regarding outcomes after mosaic embryo transfer (MET) is limited, more than 100 live births have now been documented with rather reassuring outcomes with no abnormal phenotype. Here, we aim to provide a summary of recent data regarding clinical and neonatal outcomes after transfer of mosaic embryos in IVF/PGT-A cycles.

## 1. Introduction

Preimplantation genetic testing for aneuploidy (PGT-A) of embryos during in vitro fertilization (IVF) has grown increasingly common; transfer of a euploid embryo is associated with an increased likelihood of implantation and a decreased miscarriage risk [1,2]. A basic understanding of embryologic development is needed to better understand the ongoing conversation about PGT-A and mosaicism. Embryonic genome activation does not occur until the four to eight cell (cleavage) stage; embryo development is supported by the maternally inherited messenger RNA (mRNAs) and proteins up to the cleavage stage [3,4]. Additionally, at the blastocyst stage, embryos undergo the first cellular differentiation, forming the outer trophectoderm (TE) and an inner cell mass (ICM). The TE will form the placenta, whereas the ICM will form the embryo [3]. 

The major factor leading to the failure of an embryo to result in a pregnancy or result in a miscarriage, during both natural and assisted reproductive cycles, is aneuploidy [5,6]. Most aneuploidies arise from maternal meiosis, and they increase exponentially in women over the age of 35 years, coinciding with rapidly declining IVF success and live birth rates in patients of advanced maternal age [6,7]. Research studies have shown that the incidence of aneuploidy increases from 30–50% in patients under 35 years of age to 80% in women 42 years of age or older [8]. Traditionally, morphologic assessment has been the primary technique used in prioritizing IVF embryos for transfer, but the chromosomal status of cultured embryos cannot be accurately ascertained through either static or dynamic morphologic evaluation [9,10]. PGT-A, formerly known as preimplantation genetic screening (PGS), has been proposed as a method to select IVF embryos with the highest potential of ongoing implantation based on their chromosomal make up. Although some studies have shown improved clinical outcomes with PGT-A, specifically in women with advanced maternal age, the value of PGT-A as a universal screening test for all IVF patients is yet to be determined [7,11,12]. Another potential benefit of PGT-A is the opportunity to reduce maternal and neonatal morbidity secondary to multiple gestations by allowing the transfer of fewer embryos while maintaining success rates [13]. Additionally, embryo biopsy can be done at different developmental stages of the embryo, but these stages do not all provide the same information. In recent years, with development of more physiologic culture media and improved cryopreservation techniques, there has been a worldwide shift from biopsy collection at the cleavage stage to blastocyst stage, where cells are removed from the TE [14].

The development of new diagnostic techniques for PGT-A, such as next-generation sequencing (NGS), has led to increased reporting of embryonic mosaicism. Embryonic mosaicism occurs when two or more cell populations with different genotypes are present within the same embryo [15]. They represent a third category between normal (euploid) and abnormal (aneuploid) embryos. It is also important to emphasize that in contrast to meiotically derived aneuploidy, mosaicism arises through mitotic malsegregation after fertilization and increases with cleavage-stage dysmorphism, but not with advancing maternal age [16,17]. The rate of embryonic mosaicism may vary based on a number of factors, including the stage of the embryo at the time of biopsy as well as the chromosomal detection technique used. Most initial studies involving the analysis of mosaic embryos used fluorescence in situ hybridization (FISH), a method favored because it provides information on the cytogenetic status of each cell. However, this technique has several limitations, most importantly, inability to screen for all 24 chromosomes simultaneously, hence the frequency of embryonic mosaicism after FISH varies greatly ranging from 30% to as high as 90% [17,18]. Newer molecular cytogenetic techniques such as array comparative genomic hybridization (aCGH), single nucleotide polymorphism (SNP) array, quantitative polymerase chain reaction (qPCR) and NGS were introduced to overcome FISH limitations. These advanced techniques assess the copy number of all 24 chromosomes from a single or multiple cell biopsy with ∼20–30% of blastocyst-stage embryos reported as mosaic across all maternal ages [18,19,20].

Given that a mosaic result may not necessarily represent the chromosomal constitution of the remainder of the embryo, embryos diagnosed as mosaic based on trophectoderm analysis may be fully euploid, fully aneuploid, mosaic for a euploid and an aneuploid cell line, or mosaic for two or more different abnormal cell lines.

Clinical implications of embryonic mosaicism are important in both naturally conceived and IVF pregnancies. Among natural pregnancies, mosaicism is known to affect approximately 2% of all gestations in the form of confined placental mosaicism and can lead to adverse obstetric outcome including intrauterine growth restriction (IUGR) and placental insufficiency depending on the chromosome involved [21]. In IVF embryos, most studies including a recent metanalysis have shown significantly lower implantation rate (IR), lower clinical pregnancy rate (CPR) and live birth rate (LBR) after transfer of mosaic embryos compared to those with euploid status. Compared to euploid embryos, mosaic embryo transfer (MET) has also been shown to be associated with increased risk of miscarriage in multiple studies [15,22,23]. Although information regarding neonatal outcomes is limited, now more than 100 live births after transfer of mosaic embryos have been documented with rather reassuring outcomes with no abnormal phenotype [15,16]. Here, we aim to provide a summary of recent data regarding clinical and neonatal outcomes after transfer of mosaic embryos. The PubMed and Google Scholar databases were used to search peer-reviewed publications using the following terms: ‘PGT-A’, ‘Mosaic embryos’, ‘chromosomal mosaicism’, ‘treatment outcomes’, ‘clinical outcomes’, ‘neonatal outcomes’ in combination with other keywords related to the subject area. Articles in English language that reported one or more clinical outcomes such as IR, CPR, miscarriage and/or LBR after transfer of mosaic blastocysts compared to blastocysts with euploid status were included.

## 2. Methods Applied in PGT for Chromosomal Abnormalities 

The first application of PGT was reported in 1990. A group of researchers in the UK offered PGT with transfer of only XY embryos to five couples with history of prior pregnancies affected by X-linked diseases. Amplification of a gene specific to the Y chromosome was performed using PCR after blastomere biopsy from cleavage stage embryos [24]. In 1993, FISH was used for detection of X and Y chromosomes, again in families with history of X-linked conditions. During the same year, first reports of PGT for chromosomal aneuploidy using FISH were reported [25,26]. Although cleavage stage PGT-A using FISH with transfer of normal embryos gained momentum as a part of assisted reproductive technology (ART), a major limitation of this technology was the fact that only a limited number of chromosomes, usually the most commonly affected ones, were tested. More advanced technologies use a combination of whole genome amplification (WGA) followed by analysis of all 24 chromosomes. There are about 6 picograms (pg) of DNA in a single cell. Thus, a TE biopsy of five cells provides approximately 30 pg of DNA [14]. This makes amplification a critical step since a much larger amount of DNA, typically several hundred nanograms or more, is required by technologies capable of copy number analyses. There are different methods for DNA amplification including PCR, multiple annealing and looping based amplification cycles (MALBAC) and multiple displacement amplification (MDA). There are no published studied comparing the various methods used for WGA and their predictive value in a clinical setting. Although WGA has the advantage of amplifying the entire genome, discordance as much as several thousand-fold across different portions of the genome is a limiting factor. In addition, it is important to emphasize that robust molecular techniques used for PGT-A are capable of assessing the chromosomal make up of all biopsied cells from a TE biopsy as a group, and not individually. Platforms that utilize various types of WGA for chromosomal analysis include comparative genome hybridization (CGH), arrayed CGH (aCGH) and single nucleotide polymorphism (SNP) arrays. Alternatively, quantitative or real-time PCR (qPCR) is done with targeted amplification of specific loci and NGS may be done with either. In addition, different molecular techniques used in PGT-A including various NGS platforms have different resolutions and do not detect mosaicism to the same extent [18,27]. Maxwell et al. conducted a case–control study comparing the incidence of aneuploidy and mosaicism using NGS within embryos identified as euploid by aCGH. The investigators found that of euploid embryos analyzed by aCGH, 31.6% were mosaic and 5.2% were polyploid by NGS [27]. A randomized control study comparing NGS and aCGH for preimplantation genetic screening (PGS) found that NGS was concordant with aCGH 100% of the time for 24 chromosome diagnosis and more precisely detected segmental changes when compared with aCGH [28]. In support of these findings, some studies have reported improved clinical outcomes after transfer of euploid embryos detected by NGS compared to aCGH [15,29]. More recent studies suggest that high resolution NGS (hrNGS) with Illumina’s Veriseq (Illumina Inc, Santa Clara, CA, USA) can identify chromosomal abnormalities when present in 20–80% of the TE biopsy. Therefore, in case of a TE containing five cells, this platform can identify between one and four mosaic cells [30,31]. In light of higher dynamic range and resolution compared to other methods, hrNGS has become the most commonly used method for PGT-A [32,33,34,35]. 

## 3. Is TE biopsy a Reliable Proxy for the ICM?

In case of chromosomal aneuploidy of meiotic origin, the chromosomal abnormality will be uniformly present in all cells of the preimplantation embryo. This means that the TE biopsy will reliably represent the chromosomal constitution of the embryo. This high contribution of aneuploidies with meiotic origin has led to successful application of PGT-A in ART. However, this is not the case with postzygotic mitotic errors, which result in cell lines with different karyotypes within one embryo. In the case of blastocyst stage embryos, results of a TE biopsy are used to infer the chromosomal status of the inner cell mass (ICM) and since not all cells are affected by the aneuploidy, this may lead to misdiagnosis of the embryo as aneuploid when in fact it contains a euploid ICM [36]. 

In a recent experimental study, Victor et al. reported 96.8% concordance between TE and ICM in case of whole chromosomal aneuploidy (single or multiple) in TE biopsy in 100 human blastocysts. Importantly, in case of a segmental aneuploidy in TE, this concordance dropped significantly to 42.9% with only three out of seven ICM biopsies showing aneuploidy. In this study, five out of 100 blastocysts identified as aneuploid based on their TE biopsy, had euploid ICM biopsies. Four of these samples contained segmental aneuploidies in their original TE biopsies [37]. In a similar study, Navratil et al. reported high concordance rates (~95%) between TE and the rest of the embryo (RE) biopsies. In the case of whole chromosomal aneuploidy, 59 out of 62 (95.2%) results were concordant between TE and RE. However, in the case of segmental errors there was a significant drop with only 14 out of 31 embryos (45.2%) showing concordant results between TE and RE. In the case of single whole chromosome mosaicism, 26.9% concordance was observed [38]. 

Chuang and colleagues similarly conducted a concordance analysis of TE and ICM biopsies. Thirty-three blastocysts were biopsied at three distinct locations: TE opposite to the ICM, near the ICM and within the ICM. The chromosomal status of the three different biopsy sites in the same blastocyst was determined in 29 embryos using hrNGS. The authors reported 86% (25/29) overall concordance of ploidy between TE and ICM with no remarkable difference in chromosomal ploidy between different biopsy sites. The incidence of inconsistent PGT-A results between TE and ICM defined as confined mosaicism was 14% (4/29). Out of twenty-nine embryos, one had a mosaic TE with euploid ICM (3%), one had a euploid TE with mosaic ICM (3%) and two embryos had a mosaic TE with aneuploid ICM [39]. In an earlier study, Capalbo et al. used seventy good quality human blastocysts previously identified as abnormal based on TE biopsy and CCS by aCGH. These embryos were subjected to ICM biopsy followed by nine-chromosome FISH reanalysis. In agreement with previous studies, no preferential allocation of abnormal cells between ICM and TE was observed. In addition, 97.1% (68/70) concordance was observed between ICM and TE for all chromosomes reanalyzed by FISH indicating that CCS accurately classifies embryos as diploid or aneuploid. Only two of the 50 (4%) blastocysts were mosaic, identified as aneuploid by TE biopsy and euploid by ICM reanalysis. It is important to note that the accuracy is limited due to reanalysis of only discarded embryos and limited number of chromosomes examined by FISH [40]. 


Overall, the results of these studies indicate that the findings of a TE biopsy are an excellent predictor of the chromosomal status of the entire embryo in diploid embryos or whole chromosomal aneuploidy. However, the predictive value of TE biopsy is significantly reduced in case of embryonic mosaicism including segmental abnormalities.


## 4. Considerations When Transferring Mosaic Embryos

Although it is widely accepted that mosaic embryo transfer is associated with reduced chance of implantation and increased risk of miscarriage, there is no consensus regarding specific mosaic features (level, segmental versus whole and the chromosome(s) involved) and their impact on treatment and/or pregnancy outcomes. 

Spinella et al. reported higher IR (48.9% vs. 24.2%) as well as higher CPR (40.9% vs. 15.2%) and LBR (42.2% vs. 15.2%) after transfer of mosaic embryos with <50% mosaicism compared to those with >50% mosaicism [31]. Munné and colleagues reported similar findings with higher ongoing pregnancy rate after transfer of mosaic embryos with <40% abnormal cells (50%) compared to those with 40–80% abnormal cells on the TE (27%). The ongoing pregnancy rate was even lower with transfer of complex mosaic embryos at 9% [15]. Most studies comparing clinical outcomes after transfer of mosaic embryos with segmental mosaicism and those with single whole chromosome mosaicism did not find a significant difference [15,30,41,42]. Table 1 summarizes clinical outcomes after transfer of mosaic embryos including various subtypes. 

The American Society for Reproductive Medicine (ASRM), the Preimplantation Genetic Diagnosis International Society (PGDIS), and Congress on Controversies in Preconception, Preimplantation and Prenatal Genetic Diagnosis (CoGen) have issued similar statements with recommendations for clinical management of mosaic embryos [43,45]. When no euploid embryos are available for transfer and both the clinician and patient are comfortable with transfer of a mosaic embryo, guidance exists regarding the prioritization of embryos. Based on current evidence, clinicians are encouraged to prioritize mosaic embryos with low-level mosaicism over high level mosaicism for transfer. In cases of single chromosome mosaicism, embryos mosaic for chromosomes with a known potential for IUGR (chromosomes 2, 7, 16), live born syndromes (chromosomes 13, 18, 21) or UPD should be avoided [45,46]. In a large retrospective study, Grati et al. proposed a risk scoring system for transfer of mosaic embryos based on mosaic aneuploidy detected after chorionic villus sampling (CVS) and cytogenetic analysis of products of conception [46]. The authors reported the presence of chromosomal mosaicism in 1524 (2.1%) of 72,472 CVS samples analyzed. Of the 1524 cases, 1166 underwent amniocentesis with 1011 (86%) of samples showing no signs of aneuploidy. However, 155 cases (13.3%) had TFM. 

The authors proposed classifying mosaic aneuploidies into four risk categories based on the chromosome involved and risk of fetal involvement. The investigators also detected UPD in a total of 9 cases (5.3%) with chromosomal trisomy involving chromosomes 14, 15 and 16. Based on their results, the investigators concluded that mosaic embryos with trisomy for chromosomes 1, 3, 4, 5, 6, 8, 10, 11, 12, 17, 19, 20, 22, X, Y could be considered for transfer after detailed counseling. 

## 5. Clinical and Neonatal Outcomes after Transfer of Mosaic Embryos

In a recent survey from assisted reproductive technology clinics across the united states, of the 405 clinics contacted, 252 (62.2%) responded to the survey and Ninety-one (36.1%) of them reported receiving mosaicism data on PGT-A report. Of those 91 clinics, less than half of them (42.9%) reported transferring mosaic embryos in the past [47]. Thus far, due to the limited data regarding the safety and developmental fate of mosaic embryos, most clinics are reluctant to transfer them. However, there is a rapidly growing body of evidence that suggests mosaic embryos can implant and result in healthy babies. Table 1 summarizes clinical outcomes after transfer of mosaic embryos in large clinical settings. It is important to note that some of the studies included in Table 1 are included in the compiled study conducted by Viotti and colleagues [48], which is included in Table 1.

Greco and colleagues reported clinical outcomes after transfer of mosaic embryos for the first time in 2015. Transfer of mosaic embryos was offered to 18 women who underwent IVF with no euploid embryos available for transfer. Transfer of 18 mosaic embryos resulted in eight clinical pregnancies and birth of six singletons at term. Karyotype analysis via chorionic villous sampling (CVS) showed normal karyotype in all pregnancies that progressed to term. The authors hypothesized that the extent and type of mosaicism may play a role in clinical outcomes [49]. Since then, multiple studies have been published regarding success rates with transfer of mosaic embryos. In a large study, Munne and colleagues compared pregnancy outcomes after transfer of 1327 euploid and 253 mosaic embryos in IVF-PGT-A cycles using high resolution NGS (hr-NGS). Transfer of mosiac emrbyos resulted in 37% ongoing implantation rate (OIR) and 25% miscarriage rate compared to 77% OIR and 7% miscarriage rate after transfer of euploid emrbyos.Reduced degree of mosaicism (20–40% versus > 40%) was associated with higher OIR (50% and 27%). Nevertheless, there was no difference in OIR between segmental (40%) and whole chrosome mosacisim (42%). They also reported similar OIR between mosiac monosomies and mosaic trisomies [15]. In a separate study, Munne and colleagues examined clinical outcomes after transfer of 143 mosaic embryos and reported similar outcomes with 53% IR, 24% fetal loss rate (FLR) and 41% OIR. Additionally, significanly higher FLR (24% verus 63%) and lower OIR (41% versus 63%) was seen with transfer of mosaic embryos compared to euploid embryo transfers. OIR was similar between segmental, mosaic trisomies and mosaic monosomies as well as double mosaic transfers. Complex abnotmal mosaic embryos with three or more chromosomes had significantly lower OIR (10%) (30). Zhang and colleagues reported lower IR (40.1% vs. 59%) and CPR (40.6% vs. 59.1%) after transfer of 137 mosaic embryos compared with euploid controls. However, there was no difference in IR (40.1% vs. 45.7%) or CPR (40.6% vs. 48.4%) between mosaic embryos and non-PGT controls. Four patients underwent amniocentesis after MET with no chromosomal abnormality detected. The authors also reported no difference in Birth weights between babies from MET (data available for 25 out of 36 live births) and euploid controls [23]. Victor and coleagues reported clinical outcomes after transfer of 100 blastocysts classified as mosaic by NGS. In their report, 30 clinical pregnancies resulted in 19 live births and 11 ongoing pregnancies. Non-invasive prenatal testing (NIPT) was available from seven pregnancies, all of which were normal. Results of amniocentesis were available in 11 cases. Eight of them showed normal karyotype, one case contained a balanced translocation and two cases had miscrodeletions with segments smaller than the validated resolution for their PGT-A platform [22]. In a retrospective analysis, Zhang and colleagues compared clinical and obsbtetric outcomes after transfer of 102 mosiac emrbyos after PGT-A with aCGH. Similar to other studies, MET was associated with lower biochemical (65.7% vs. 76.1%) and LBR (46.6% vs. 59/1%) compared with euploid embryo transfer. Although transfer of euploid embryos was associated with higher LBR compared to MET with segmental mosaicism, it did not reach statistical significance (48.3% vs. 59.1%). The authors also reported perinatal outcomes in 48 live births after MET. No difference in birth weight, preterm delivery rate, or risk of congenital malformations was detected after MET compared with euploid emrbryos. Three patients had prenatal diagnostic testing with amniocentesis after MET which yeilded normal karyotypes [41]. Table 2 summarizes prenatal diagnostic/screening test results as well as neonatal outcomes after transfer of mosaic embryos.

## 6. Conclusions

Growing evidence suggests that MET is associated with lower implantation rate and higher risk of miscarriage compared with euploid embryo transfer. However, the data regarding whether the level and/or the type (segmental, whole chrosmosome or complex) of mosaicisim play a significant role in treatment outcomes are conflicting. Existing data regarding neonatal outcomes after MET are somewhat reassuring, with LBR ranging from 30% to 48% and miscarriage rates ranging from 20% to 33% per mosaic embryo transfer. However, many questions remain unanswered such as the reliability of prenatal screening methods such as cell-free fetal DNA tests (NIPT), risk of congenital abnormalities, and long term outcomes of infants born after MET. Most experts agree that transfer of mosaic embryos should only be considered in situations in which no euploid embryos are available for transfer and after comprehensive genetic counseling with an emphasis on prenatal diagnostic testing (CVS or amniocentesis) and discussion of alternative options including third party reproduction. Future studies that focus on perinatal and long-term outcomes of children born after transfer of mosaic emryos may help elucidate the potential long-term implications of MET.

## Figures and Tables

**Table 1 jcm-10-01369-t001:** Clinical outcomes after mosaic embryo transfer (MET).

						Mosaic vs. Euploid				
Author	Year	Design	Event	Control	Method of Detection	IR	CPR	MR	OPR	LBR
Munne [30]	2017	Retrospective	143 Mosaic Blastocysts	1045 Euploid Blastocysts	NGS	53% vs. 71% (*p* > 0.05)	40% vs. 63% (*p* = 0.006)	25% vs. 10% (*p* = 0.002)		
Munne [15]	2020	Retrospective	253 Mosaic Blastocysts	2654 Euploid Blastocysts	NGS	49% vs. 83% (*p* < 0.001)	37% vs. 77% (*p* < 0.001)	25% vs. 7% (*p* < 0.001)	37% vs. 85% (*p* < 0.001)	
Zhang [23]	2020	Prospective	137 Mosaic Blastocysts	476 Euploid Blastocysts	NGS	40.1% vs. 59% (*p* < 0.001)	40.6% vs. 59.1% (*p* < 0.05)	33.3% vs. 20.5% (*p* = 0.05%)	27.1% vs. 47% (*p* < 0.001)	
Zhang [41]	2019	Retrospective	102 Mosaic Blastocysts	268 Euploid Blastocysts	aCGH		57.8% vs. 67.5% (*p* = 0.08)	20.3% vs. 12.7% (*p* = 0.15)		46.6% vs. 59.1% (*p* = 0.03)
Lee [42]	2020	Retrospective	83 Mosaic Blastocysts	216 Euploid Blastocysts	NGS	51.8% vs. 65.7% (*p* < 0.05)			47% vs. 64.8% (*p* < 0.05)	
Spinella [31]	2018	Prospective	77 Mosaic Blastocysts	251 Euploid blastocysts	NGS or aCGH	38.5% vs. 54.6% (*p* = 0.02)			30% vs. 46.4% (*p* = 0.019)	30% vs. 46.4% (*p* = 0.01)
Victor [22]	2019	Prospective	100 Mosaic Blastocysts	478 Euploid Blastocysts	NGS	38% vs. 49.6% (*p* = 0.02)	30% vs. 47.1% (*p* = 0.001)			
Viotti [43]	2020	Retrospective	1000 Mosaic Blastocysts	5561 Euploid Blastocysts	NGS	46.5% vs. 57.2% (*p* < 0.05)		20.4% vs. 8.6%	37% vs. 52.3% (*p* < 0.05)	
						Segmental vs. Whole chromosome (Single Aneuploid)				
Author	Year	Design	Event	Control	Method of Detection	IR	CPR	MR	OPR	LBR
Munne [30]	2017	Retrospective	143 Mosaic Blastocysts	1045 Euploid Blastocysts	NGS		41% vs. 50% (*p* > 0.05)			
Munne [15]	2020	Retrospective	253 Mosaic Blastocysts	2654 Euploid Blastocysts	NGS		40% vs. 42% (*p* > 0.05)			
Zhang [41]	2019	Retrospective	102 Mosaic Blastocysts	268 Euploid Blastocysts	aCGH		53.6% vs. 54.8% (*p* = 0.06)	13.3% vs. 20.6% (*p* = 0.22)		48.3% vs. 43.5% (*p* = 0.02)
Lee [42]	2020	Retrospective	83 Mosaic Blastocysts	216 Euploid Blastocysts	NGS				46.7% vs. 45.5% (*p* > 0.05)	
						Segmental vs. Complex chromosomal				
Author	Year	Design	Event	Control	Method of Detection	IR	CPR	MR	OPR	LBR
Munne [30]	2017	Retrospective	143 Mosaic Blastocysts	1045 Euploid Blastocysts	NGS		41% vs. 10% (*p* < 0.001)			
Munne [15]	2020	Retrospective	253 Mosaic Blastocysts	2654 Euploid Blastocysts	NGS		40% vs. 9% (*p* < 0.001)			
Lee [42]	2020	Retrospective	83 Mosaic Blastocysts	216 Euploid Blastocysts	NGS				46.7% vs. 45.5% (*p* > 0.05)	
Viotti [43]	2020	Retrospective	1000 Mosaic Embryos	5561 Euploid Blastocysts	NGS	51.6% vs. 43.1% (*p* < 0.05)			43.1% vs. 34.7% (*p* < 0.05)	
						Segmental vs. Euploid				
Author	Year	Design	Event	Control	Method of Detection	IR	CPR	MR	OPR	LBR
Victor [22]	2019	Prospective	100 Mosaic Blastocysts	478 Euploid Blastocysts	NGS	Single Segmental				
						45.5% vs. 49% (*p* = 0.7)	39.4% vs. 47.1% (*p* = 0.4)			
						Multiple Segmental				
						36.4% vs. 49% (*p* = 0.5)	27.3% vs. 47.1% (*p* = 0.2)			
						Whole Chromosome vs. Euploid				
Author	Year	Design	Event	Control	Method of Detection	IR	CPR	MR	OPR	LBR
Victor [22]	2019	Prospective	100 Mosaic Blastocysts	478 Euploid Blastocysts	NGS	27.9% vs. 49% (*p* = 0.006)	23.3% vs. 47.1% (*p* = 0.003)			
Viotti [43]	2020	Retrospective	1000 Mosaic Blastocysts	5561 Euploid Blastocysts	NGS	41.8% vs. 57.2%			31.3% vs. 52.3%	
						Low Level Mosaicism (<50%) vs. High level Mosaicism (>50%)				
Author	Year	Design	Event	Control	Method of Detection	IR	CPR	MR	OPR	LBR
Lin [44]	2020	Retrospective	108 Mosaic Blastocysts	None	NGS	51.8% vs. 52% (*p* = 0.98)	47% vs. 52% (*p* = 0.66)	5.1% vs. 30.7% (*p* = 0.01)	47% vs. 36% (*p* = 0.33)	44.6% vs. 36% (*p* = 0.45)
Spinella [31]	2018	Prospective	77 Mosaic Blastocysts	251 Euploid Blastocysts	NGS or aCGH	48.9% vs. 24.2 (*p* = 0.039)			40.9% vs. 15.2% (*p* = 0.02)	42.2% vs. 15.2% (*p* = 0.02)
Munne [15]	2020	Retrospective	253 Mosaic Blastocysts	2654 Euploid Blastocysts	NGS		50% vs. 27% (*p* < 0.02)			
Viotti [43]	2020	Retrospective	1000 Mosaic Blastocyst	5561 Euploid Blastocysts	NGS	47.4% vs. 35.8% (*p* < 0.05)			39.5% vs. 24.1% (*p* < 0.05)	

Note: IR = implantation rate, CPR = clinical pregnancy rate, MR = miscarriage rate, ORP = ongoing pregnancy rate, LBR = live birth rate.

**Table 2 jcm-10-01369-t002:** Prenatal diagnostic/screening results and neonatal outcomes after MET.

Author	Year	Event	Antenatal Testing		Antenatal Testing if Abnormal	Birth Weight (Grams)		Neonatal Outcomes
			Normal Result on NIPT	Normal Karyotype on Amniocentesis or CVS		Mosaic	Euploid	
Zhang [23]	2020	137 Mosaic Blastocysts	4/4	1/1		3180 ± 505 g (*n* = 24)	3047 ± 560 g (*n* = 64)	
Zhang [41]	2019	102 Mosaic Blastocysts		3/3		3038 ± 0.65 (*n* = 48)	3030 ± 0.50 (*n* = 159)	No Congenital anomalies
Lee [42]	2020	83 Mosaic Blastocysts		40/40		2997.7 ± 501.1 (*n* = 36)	3146.2 ± 450.0 (*n* = 120)	No congenital anomalies, No difference in gestational age at delivery
Spinella [31]	2018	77 Mosaic Blastocysts		24/24				All 24 ongoing clinical pregnancies went to term with Live born Infants
Victor [22]	2019	100 Mosaic Blastocysts	7/7	8/11	1 case contained a balanced translocation, and two cases showed microdeletions affecting segments smaller than the validated resolution of the PGT-A platform used			19 live birth and 11 ongoing pregnancies after MET
Greco [49]	2015	18 Mosaic Blastocysts		6/6				All 6 pregnancies resulted in delivery of healthy infants
Hong [50]	2020	28 Mosaic Blastocysts		3/4	1 case with balanced translocation			All 5 pregnancies resulted in delivery of healthy infants

Note: NIPT = Non-Invasive prenatal testing, CVS = Chorionic villous sampling.

## Data Availability

The authors confirm that the data supporting this review are available within the article and the referrence list provided.

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
