# Peer review of "Pregnancy and Neonatal Outcomes after Transfer of Mosaic Embryos: A Review"

_jcm, 2021, doi:10.3390/jcm10071369_

Round 1

Reviewer 1 Report

The topic of the article with the title „Pregancy and neonatal outcomes after transfer of mosaic embryos: a review” is within the scope of the “Journal of clinical medicine”.

Preimplantation genetic testing for aneuploidy with next generation sequence (NGS) is a modern aspect of reproductive medicine. Several studies and recommendations are published worldwide.

Nevertheless, the question of clinical impact of this genetic testing of embryos and the relevance of mosaics for clinical work is not clearly answered. This review tries to give an overview on the existing studies and data about treatment and neonatal outcome after transfer of mosaic embryos.

After an interestingly written introduction to the topic the historical development of PGS is described. The method/technique of literature search and review is not provided. The main part of the article with discussion of the technique seems to be partly unstructured. The tables are informative, include important data and are well structured.

Literature is up to date and quoted adequately.

Language: very elaborate style.

Some sentences are difficult to be understood on the first glance:

E.g. Line 27

A basic understanding of embryological development frames the conversation about PGT-A and mosaicism.

= A basic understanding of embryological development is needed to understand the ongoing discussion about PGT-A and mosaicism.

Minor comments:

Line 96 – In

It may be assumed, that the bold text passages are parts of correction – or with intention to highlight important sentences?

There are about 6 picogram (pg) of DNA in a single cell. – a statement like this and ff. should be quoted

Many spelling mistakes in Conclusion: Line 261 associated, 266 neonatal, 268 unanswered, 272 comprehensive

Author Response

Journal of Clinical Medicine

Dear Editorial Board Members,

We are very grateful for the constructive comments from the Reviewers on our manuscript, “Pregnancy and Neonatal Outcomes after Transfer of Mosaic Embryos: A Review”. We have made all of the suggested changes and included in line-by-line response below.

The topic of the article with the title „Pregnancy and neonatal outcomes after transfer of mosaic embryos: a review” is within the scope of the “Journal of clinical medicine”.

Preimplantation genetic testing for aneuploidy with next generation sequence (NGS) is a modern aspect of reproductive medicine. Several studies and recommendations are published worldwide.

Nevertheless, the question of clinical impact of this genetic testing of embryos and the relevance of mosaics for clinical work is not clearly answered. This review tries to give an overview on the existing studies and data about treatment and neonatal outcome after transfer of mosaic embryos.

After an interestingly written introduction to the topic the historical development of PGS is described. The method/technique of literature search and review is not provided. The main part of the article with discussion of the technique seems to be partly unstructured. The tables are informative, include important data and are well structured.

Thank you for your comments and suggestions. The manuscript has been revised and modified to reflect the points discussed above.

“Here we aim to provide a summary of recent data regarding clinical and neonatal outcomes after transfer of mosaic embryos. The PubMed and Google Scholar databases were used to search peer-reviewed publications using the following terms: ‘PGT-A’, ‘Mosaic embryos’, ‘chromosomal mosaicism’, ‘treatment outcomes’, ‘neonatal outcomes’ in combination with other keywords related to the subject area. Relevant articles in English language that reported one or more treatment outcomes such as implantation, clinical pregnancy, miscarriage and/or LBR after transfer of mosaic blastocysts compared to blastocysts with euploid status were included.” (lines 97-104)

“Maxwell et al conducted a case-control study comparing the incidence of aneuploidy and mosaicism using NGS within embryos identified as euploid by aCGH. The investigators found that of euploid embryos analyzed by aCGH, 31.6% were mosaic and 5.2% were polyploid by NGS. A randomized control study comparing NGS and aCGH for preimplantation genetic screening (PGS) found that NGS was concordant with aCGH 100% of the time for 24 chromosome diagnosis and more precisely detected segmental changes when compared with aCGH. In support of these findings, some studies have reported improved treatment outcomes after transfer of euploid embryos detected by NGS compared to aCGH.” (lines 133-141)

Literature is up to date and quoted adequately.

Thank you.

Language: very elaborate style.

Thank you.

A basic understanding of embryological development frames the conversation about PGT-A and mosaicism.

= A basic understanding of embryological development is needed to understand the ongoing discussion about PGT-A and mosaicism.

The sentence has been modified accordingly.

“A basic understanding of embryologic development is needed to better understand the ongoing conversation about PGT-A and mosaicism.” (lines 36-37)

Line 96 – In

Corrected.

It may be assumed, that the bold text passages are parts of correction – or with intention to highlight important sentences?

Thank you. The authors did not mean for any part of the text to be in bold. It has been revised and corrected.

Many spelling mistakes in Conclusion: Line 261 associated, 266 neonatal, 268 unanswered, 272 comprehensives

Thank you. The manuscript has been revised with spelling and grammatical mistakes corrected.

Reviewer 2 Report

The manuscript is well-written and the topic is discussed adequately.  The results confirm the objective of the topic. The reference titles are appropriately cited.

Author Response

Journal of Clinical Medicine

Dear Editorial Board Members,

We are very grateful for the constructive comments from the Reviewers on our manuscript, “Pregnancy and Neonatal Outcomes after Transfer of Mosaic Embryos: A Review”. We have made all of the suggested changes and included in line-by-line response below.

The manuscript is well-written, and the topic is discussed adequately.  The results confirm the objective of the topic. The reference titles are appropriately cited.

Thank you!

Reviewer 3 Report

In this paper, the authors aimed to provide a summary of recent data regarding clinical and neonatal  outcomes after transfer of mosaic embryos in IVF/PGT-A cycles. They also considered several points of interest regarding mosaic embryo: the methods of the mosaicism detection, the TE-ICM concordance, the recommendations for clinical management of MET.

Although a recent meta-analysis analyzed the pregnancy outcomes after MET [Zhang 2020; Genes:11(9)], Abhari and Kawwass assessed the data comparing different types of aneuploidy (complex, segmantal….) and levels of mosaicism (low and high). They also evaluated the neonatal outcomes.

There are some issues that need to be addressed.

  1. Others papers describe the outcomes after MET (for examples, Fragouli, E. et al. Hum. Genet. 2017;136:805–819; Lledó, B. et al. Syst. Biol. Reprod. Med. 2017;63:1-3…..). Please explain the selection criteria used by the authors.

  1. Modify the paragraph “2.Methods applied in PGT for Chromosomal Abnormalities”.

This issue has been well described in the review of Viotti 2020 (ref [14]). In this context, the authors should argue about the specificity and sensitivity of the techniques in the mosaicism detection.

  1. Revise the paragraph “Considerations when transferring mosaic embryos”.

Also this topic has been well described in the review of Viotti 2020 (ref [14]). In this paper, the authors should discuss about the transfer of MET considering the percentage of mosaicism and the type of aneuploidy (segmental, complex…..).

  1. For greater understanding, split the paragraph “5.Treatment and Neonatal Outcomes after Transfer of Mosaic Embryos” in two parts:
    • the clinical/pregnancy outcomes.
    • the neonatal outcomes. 

In the first part, the authors should highlight the outcomes divided by the type of aneuploidy    (complex, segmental) and percentage of mosaicism. In the second one, the authors should carefully describe the neonatal outcomes.

  1. In Viotti’s paper, some results were from previously published papers (Munne 2017, Munne 2020, Spinella 2018, Victor 2019). It’s important to discuss this point.

  1. Please modify the text:
    1. LINE 86: It is necessary to comment the results of the recent metanalysis on the pregnancy outcomes after MET (Zhang 2020, Genes ref. [23]).
    2. change “IVF treatment outcome” in “clinical/pregnancy outcomes” in the text and in the tables.
    3. LINE 199: Change “treatment” in “clinical/pregnancy”
    4. LINE 217: Insert the reference number [45]
    5. LINE 228: Correct “implantation”
    6. LINE 240 and LINE 251: please, do not change paragraph when discussing the same paper.
    7. LINE 249: please, clarify “in a subgroup analysis”.

  1. There are two reference 44. Please, correct the reference number of Viotti 2020 in the text and in the tables.

Author Response

Journal of Clinical Medicine

Dear Editorial Board Members,

We are very grateful for the constructive comments from the Reviewers on our manuscript, “Pregnancy and Neonatal Outcomes after Transfer of Mosaic Embryos: A Review”. We have made all of the suggested changes and included in line-by-line response below.

Other papers describe the outcomes after MET (for examples, Fragouli, E. et al. Hum. Genet. 2017;136:805–819; Lledó, B. et al. Syst. Biol. Reprod. Med. 2017;63:1-3…..). Please explain the selection criteria used by the authors.

Thank you for the comment. The introduction section has been modified to reflect selection criteria:

“Here we aim to provide a summary of recent data regarding clinical and neonatal outcomes after transfer of mosaic embryos. The PubMed and Google Scholar databases were used to search peer-reviewed publications using the following terms: ‘PGT-A’, ‘Mosaic embryos’, ‘chromosomal mosaicism’, ‘treatment outcomes’, ‘neonatal outcomes’ in combination with other keywords related to the subject area. Relevant articles in English language that reported one or more treatment outcomes such as implantation, clinical pregnancy, miscarriage and/or LBR after transfer of mosaic blastocysts compared to blastocysts with euploid status were included.” (lines 97-104)

Modify the paragraph “2. Methods applied in PGT for Chromosomal Abnormalities”.

This issue has been well described in the review of Viotti 2020 (ref [14]). In this context, the authors should argue about the specificity and sensitivity of the techniques in the mosaicism detection

Thank you for the comment. The following section has been added to the manuscript:

“Maxwell et al conducted a case-control study comparing the incidence of aneuploidy and mosaicism using NGS within embryos identified as euploid by aCGH. The investigators found that of euploid embryos analyzed by aCGH, 31.6% were mosaic and 5.2% were polyploid by NGS (27). A randomized control study comparing NGS and aCGH for preimplantation genetic screening (PGS) found that NGS was concordant with aCGH 100% of the time for 24 chromosome diagnosis and more precisely detected segmental changes when compared with aCGH (28). In support of these findings, some studies have reported improved treatment outcomes after transfer of euploid embryos detected by NGS compared to aCGH.” (lines 133-141)

Revise the paragraph “Considerations when transferring mosaic embryos”.

Also, this topic has been well described in the review of Viotti 2020 (ref [14]). In this paper, the authors should discuss about the transfer of MET considering the percentage of mosaicism and the type of aneuploidy (segmental, complex…..).

Thank you. The following section has been added to the manuscript based on the expert recommendation:

“Spinella et al reported higher implantation rate (48.9% vs 24.2%) as well as higher clinical pregnancy (40.9% vs 15.2%) and live birth rate (42.2% vs 15.2%) after transfer of mosaic embryos with <50% mosaicism compared to those with >50% mosaicism (31). Munné and colleagues reported similar findings with higher ongoing pregnancy rate after transfer of mosaic embryos with <40% abnormal cells (50%) compared to those with 40-80% abnormal cells on the TE (27%). Ongoing pregnancy rate was even lower with transfer of complex mosaic embryos at 9% (15). Most studies comparing clinical outcomes after transfer of mosaic embryos with segmental mosaicism versus those with single whole chromosome mosaicism did not find a significant difference (15, 30, 41, 42). Table 1 summarizes clinical outcomes after transfer of mosaic embryos including various subtypes. “(lines 197-205)

For greater understanding, split the paragraph “5. Treatment and Neonatal Outcomes after Transfer of Mosaic Embryos” in two parts:

  • the clinical/pregnancy outcomes.
  • the neonatal outcomes

Thank you for the comment. The authors are in complete agreement with the reviewer’s suggestion and have provided a summary of clinical and neonatal outcomes in two separate tables. With all due respect, the authors believe that the manuscript with its current structure provides a reasonable flow for readers regarding clinical and birth/neonatal outcomes from the same cohort of patients.

In Viotti’s paper, some results were from previously published papers (Munne 2017, Munne 2020, Spinella 2018, Victor 2019). It’s important to discuss this point.

Thank you for the comment. The following section has been added to the manuscript to reflect this point:

“It is important to note that some of the studies included in table 1 regarding clinical outcomes after MET are also included in the compiled study conducted by Viotti and colleagues (49), which has also been included in table 1”. (lines 235-237)

Please modify the text:

LINE 86: It is necessary to comment the results of the recent metanalysis on the pregnancy outcomes after MET (Zhang 2020, Genes ref. [23]).

Thank you. The manuscript was modified to include the findings of the metanalysis as well:

“In IVF embryos, most studies including a recent metanalysis have shown significantly lower implantation rate (IR), lower clinical pregnancy rate (CPR) and live birth rate (LBR) after transfer of mosaic embryos compared to those with euploid status. Compared to euploid embryos, MET has also been shown to be associated with increased risk of miscarriage in multiple studies”. (lines 91-95)

Change “IVF treatment outcome” in “clinical/pregnancy outcomes” in the text and in the tables.

Thank you. The manuscript has been modified as recommended 

LINE 199: Change “treatment” in “clinical/pregnancy”

 Thank you. The manuscript has been modified as recommended 

LINE 217: Insert the reference number [45]

Reference has been added. 

LINE 228: Correct “implantation”

Corrected. Thank you!

LINE 240 and LINE 251: please, do not change paragraph when discussing the same paper.

Corrected. Thank you!

LINE 249: please, clarify “in a subgroup analysis

 To further clarify this, the section was changed to the following:

“Although transfer of euploid embryos was associated with higher LBR compared to MET with segmental mosaicism,  it did not reach statistical significance (48.3% vs.59.1%).” (lines 283-285)

Round 2

Reviewer 3 Report

Thank you for editing the text as required. 

Please clarify "relevant article" (line 101) indicating the selection criteria.

After this change, the manuscript deserves to be published.

Author Response

Dear Editorial Board Members,

We are very grateful for the constructive comments from the Reviewers on our manuscript, “Pregnancy and Neonatal Outcomes after Transfer of Mosaic Embryos: A Review”. We have made all of the suggested changes and included in line-by-line response below.

Please clarify "relevant article" (line 101) indicating the selection criteria.

Thank you. The manuscript has been modified to address this point.

“The PubMed and Google Scholar databases were used to search peer-reviewed publications using the following terms: ‘PGT-A’, ‘Mosaic embryos’, ‘chromosomal mosaicism’, ‘treatment outcomes’, ‘clinical outcomes’, ‘neonatal outcomes’ in combination with other keywords related to the subject area. Articles in English language that reported one or more clinical outcomes such as IR, CPR, miscarriage and/or LBR after transfer of mosaic blastocysts compared to blastocysts with euploid status were included.” (Lines 99-104)